# Inflammaging and Vascular Function in Metabolic Syndrome: The Role of Hyperuricemia

**DOI:** 10.3390/medicina58030373

**Published:** 2022-03-02

**Authors:** Agnė Laučytė-Cibulskienė, Monika Smaliukaitė, Jolanta Dadonienė, Alma Čypienė, Jurgita Mikolaitytė, Ligita Ryliškytė, Aleksandras Laucevičius, Jolita Badarienė

**Affiliations:** 1Department of Clinical Sciences Malmo, Lund University, Skane University Hospital, 205 02 Malmö, Sweden; agne.laucyte-cibulskiene@med.lu.se; 2Faculty of Medicine, Vilnius University, 03101 Vilnius, Lithuania; jolanta.dadoniene@mf.vu.lt (J.D.); alma.cypiene@santa.lt (A.Č.); ligita.ryliskyte@santa.lt (L.R.); aleksandras.laucevicius@santa.lt (A.L.); jolita.badariene@santa.lt (J.B.); 3State Research Institute Centre for Innovative Medicine, 08406 Vilnius, Lithuania; jurgita.mikolaityte@santa.lt

**Keywords:** arterial stiffness, carotid-femoral pulse wave velocity, hyperuricemia, inflammaging, metabolic syndrome, uric acid

## Abstract

*Background and Objectives*: Early vascular aging determines a more rapid course of age-related arterial changes. It may be induced by a proinflammatory state, caused by hyperuricemia and metabolic syndrome and their interrelationship. However, the impact of serum uric acid (SUA) on early arterial stiffening and vascular function remains uncertain. *Materials and Methods*: A total of 696 participants (439 women aged 50–65 and 257 men aged 40–55) from the Lithuanian High Cardiovascular Risk (LitHiR) primary prevention program were enrolled in the study. They underwent anthropometric measurements and laboratory testing along with arterial parameters’ evaluation. Quality carotid stiffness (QCS), carotid-radial pulse wave velocity (crPWV), carotid-femoral pulse wave velocity (cfPWV), flow-mediated dilatation (FMD), and carotid intima-media thickness (CIMT) were registered. *Results*: We found that hyperuricemia was significantly associated with inflammation, registered by high-sensitivity C-reactive protein in both sexes. A very weak but significant association was observed between cfPWV and SUA in men and in women, while, after adjusting for risk factors, it remained significant only in women. A positive, weak, but significant association was also observed for QCS, both right and left in women. No relationship was observed between crPWV, FMD, CIMT, and SUA.

## 1. Introduction

The concept of early vascular aging (EVA) describes individuals with accelerated age-related vascular remodeling and dysfunction compared to expected aging. EVA has been shown to be responsible for a number of cardiovascular complications, including coronary heart disease, cognitive decline and dementia, chronic kidney disease, periphery artery disease, and stroke [1]. 

The recent concept of inflammaging is understood as an age-related condition observed when sterile, low-grade, chronic inflammation is present [2,3]. It explains why patients with rheumatoid arthritis, inflammatory bowel disease, uremia [3], and other conditions related to persistent inflammation, e.g., metabolic syndrome [4], present increased arterial stiffness [5] and, thereby, higher cardiovascular risk [2]. 

Metabolic syndrome, on the other hand, remains significant in today’s world. It is considered as a proinflammatory state; it, thus, may be associated with EVA and, therefore, with increased cardiovascular disease (CVD) risk [6]. Oxidative stress is closely associated with metabolic syndrome; more precisely, obesity and insulin resistance are the key factors leading to it [7]. At the onset of the syndrome, toll-like receptors (TLR) and proinflammatory nuclear factor-kappa B (NF-κB) are activated; this contributes to the synthesis of other proinflammatory molecules (tumor necrosis factor-α, interleukine-6, chemokines) and, therefore, accelerates EVA [8,9]. A recent study has also shown that reducing overweight with moderate calorie restrictions not only reduces the inflammation process, measured by high-sensitivity C-reactive protein (hs-CRP) and other inflammatory markers, but also improves antioxidant capacity in individuals and improves glucose tolerance [10].

Hyperuricemia is also associated with metabolic syndrome, as the relationship has been confirmed by previous cross-sectional studies [11]; however, the possible pathophysiological mechanisms are not fully understood. Uric acid, a final compound of the complex reactions in the purines’ metabolism, could be a possible trigger for vascular dysfunction [12]. Hyperuricemia might induce endothelial dysfunction [13] and the expression of hepatic inflammatory molecules by activating the proinflammatory NF-κB transcriptional factors [12] and contribute to accelerated vascular aging [14]. In physiological concentrations, uric acid acts as an antioxidant; however, when uric acid levels exceed the normal range, they affect proinflammatory arterial stiffening [13]. Studies have shown a strong association between hyperuricemia and CVD, but the independent modulatory role of uric acid on the arterial stiffening process remains unclear [13,15,16]. 

In this study, we hypothesized that a hyperuricemia-caused proinflammatory phenotype in the metabolic syndrome contributes to EVA, measured by different parameters such as arterial stiffness (carotid-femoral pulse wave velocity (cfPWV), carotid-radial pulse wave velocity (crPWV), and quality carotid stiffness (QCS)), carotid-intima media thickness (CIMT), and endothelial function, measured by flow-mediated dilatation (FMD). This article aims to establish the importance of hyperuricemia’s role on vascular health in the metabolic syndrome population.

## 2. Material and Methods

### 2.1. Study Population

This study was carried out at Vilnius University Hospital Santaros klinikos Preventive Cardiology sub-department between January 2018 and November 2019. Permission for the study was obtained from the Vilnius Regional Committee of Bioethics (No. 158200-18/4-1006-521). A total of 710 attendees were selected from the Lithuanian High Cardiovascular Risk (LitHiR) primary prevention program [17], including women aged from 50 to 65 and men aged from 40 to 55 years old with diagnosed metabolic syndrome in line with the 2005 National Cholesterol Education Program Adult Treatment Panel III (NCEP ATP III) modified criteria [18]. For risk assessment for cardiovascular diseases for this age range, we used the European Society of Cardiology Guidelines on Cardiovascular disease prevention in clinical practice [19]. Age range by sexes differed by reason: 60-year-old women have a similar risk for CVD as 50-year-old men [19]. Exclusion criteria were diagnosed CVD and usage of xanthine oxidase inhibitors. Subjects with diagnosed gout who did not declare the usage of xanthine oxidase inhibitors were included. 

### 2.2. Assessment of Study Population

After signing the informed consent form all subjects underwent anthropometric evaluation of height and weight. Body mass index (BMI) was calculated using the formula BMI = weight/height^2^ (kg/m^2^). Blood pressure was measured using a manual sphygmomanometer (Riester precisa^®^ N Sphygmomanometer, Germany). A cuff was positioned on the upper arm while sitting, according to all recommendations from the European Society of Cardiology Guidelines for the management of arterial hypertension [20]. Mean arterial pressure (MAP) was calculated using the formula MAP = 1/3 systolic blood pressure + 2/3 diastolic blood pressure [21]. Blood samples for serum uric acid (SUA) concentration, fasting plasma glucose (FPG), total cholesterol (TC), high-density lipoprotein (HDL) cholesterol, low-density lipoprotein (LDL) cholesterol, triglycerides (TG), serum creatinine, and hs-CRP were analyzed using the “Abbott Architect ci8200 PLUS” (Abbott Laboratories, USA) analyzer. The estimated glomerular filtration rate (eGFR) was calculated using the CKD-EPI equation [22]. Attendees completed a questionnaire about smoking. The results were divided into two groups: smokers (current and former smokers) and non-smokers (never-smokers). Hyperuricemia was diagnosed when the SUA levels in the blood serum exceeded 357 µmol/L in women and 428 µmol/L in men.

Clinical examination and assessment of non-invasive arterial parameters were performed in a quiet room with a comfortable temperature (22–24 °C) in the morning. Participants were asked not to eat and drink alcohol or caffeinated drinks at least 12 h before the lab tests and arterial stiffness assessment. 

### 2.3. Non-Invasive Assessment of Arterial Stiffness 

#### 2.3.1. Assessment of Carotid Artery Intima-Media Thickness (CIMT) and Quality Carotid Stiffness (QCS)

An Art.Lab (Esaote Europe B.V., Philipsweg 1, 6227 AJ Maastricht, The Netherlands) system was used to assess the carotid wall. Intima-media thickness was measured six times in an area no shorter than 15 mm in the common carotid artery. To get the derivate QCS, B mode and an arterial wall motion program were used. Systolic and diastolic blood pressures were measured together with the assessment of the carotid artery cross-sectional area during systole and diastole. Six measurements of QCS with its mean and standard deviation were provided on the screen. 

#### 2.3.2. Assessment of Pulse Wave Velocity (PWV) 

Carotid and radial pulse waves were evaluated in the supine position in a non-invasive way by using the tonometry technique (Sphygmocor (v.7.01) AtCor Medical Pty. Ltd. 1999–2002, Sydney, Australia) while simultaneously recording ECG. Pulse wave velocity (PWV) was determined by measuring the carotid-to-radial (crPWV) and carotid-to-femoral (cfPWV) pulse wave transit times. 

#### 2.3.3. Assessment of Flow-Mediated Dilatation (FMD)

FMD assessment was performed using a technique described in the expert consensus and evidence-based recommendations for the assessment of flow-mediated dilation in humans [23]. Patients were asked to lay down on the examination table comfortably. The brachial artery was scanned longitudinally above the antecubital fossa. The arterial diameter was assessed at baseline by making 10–15 scans at the end of diastole, measuring the space between the intima/lumen interface. To measure the brachial artery during reactive hyperemia, a pneumatic tourniquet was inflated >50 mmHg above systolic pressure for 5 min to stop arterial flow. Then, the cuff was deflated and longitudinal scans were obtained continuously for up to 3 min. Software programs (CVI Acquisition, Vascular Analysis Tools, Vascular Converter CVI, and Brachial Analyzer, Medical Imaging Application, 1998–2003 LLC, Iowa City, IA 52246, USA) were used for this test, as well as the ECG being recorded simultaneously.

### 2.4. Statistical Analysis

Analyses were performed using R-Commander (Version 4.0.5). The assumption of normality was verified by visualizing each variable where the symmetrical histogram of the variable was considered as a normal distribution, while variables with obvious deviations were considered as not normally distributed and were summarized as median and the range. Normally distributed continuous data were indicated as mean ± standard deviation. Discrete variables were presented in percentage (number). In order to present sociodemographic and clinical data according to sex, the Welch Two Sample t-test (for normally distributed data) and the Wilcoxon rank sum test (for non-normally distributed data) were completed. To make a detailed analysis, the subjects were divided into sex-specific quartiles (Q1, Q2, Q3, Q4) according to the SUA concentration in order to reveal the association between SUA and various evaluated parameters. An ANOVA model was performed for normally distributed data and the Kruskal–Wallis test for non-normally distributed data. Additionally, the Spearman rank order was used to identify the correlation between the variables and the SUA levels in the blood. Finally, multiple linear regression analysis was applied with cfPWV as a dependent variable. A *p*-value < 0.05 was considered as significant.

## 3. Results

### 3.1. Baseline and Vascular Characteristics

This cross-sectional study included 710 patients, of whom 14 were excluded due to usage of xanthine oxidase inhibitors. In total, 696 patients (439 females (63%) and 257 males (37%)) who signed the informed consent were enrolled. Of those, 34% had hyperuricemia and 2% had gout. We found sex specific differences in descriptive characteristics (Table 1). Women had higher BMI and LDL cholesterol and lower eGFR, while men had lower HDL cholesterol, higher TG, SUA, and MAP, and smoking prevalence. FPG and the inflammation process, defined by hs-CRP, had no significant difference between the sexes.

Vascular parameters, except for crPWV, were significantly higher in women (Table 2). No significant difference in respect to sex was observed in FMD. 

### 3.2. Association between Objective Data, Vascular Parameters, and SUA

To reveal the association between hyperuricemia and vascular variables, the SUA levels were divided into sex-specific quartiles (Table 3 and Table 4, Figure 1). The first (Q1) and second (Q2) quartiles in men and women covered normal SUA levels and the fourth (Q4) represented hyperuricemia, while the third (Q3) included subjects with SUA concentration that was near the upper normal limit or slightly over it. Both men and women in the fourth quartile (Q4) as compared to the first (Q1) were fatter (higher BMI), had higher TG and HDL cholesterol levels, lower eGFR, and higher hs-CRP while MAP and FPG were elevated only in women. There was no age difference in women, but men with hyperuricemia were younger. 

Furthermore, cfPWV significantly increased as the SUA concentration increased, reaching its maximum in the fourth quartile (Q4) (Q4, 9.06 in women and 8.03 in men; Q1, 8.36 and 7.87, respectively). In addition, in women, QCS was also significantly higher in those with a higher SUA level (Figure 1).

The Spearman’s correlation between SUA concentration and descriptive and vascular stiffness is presented in Table 5. Both in men and women, SUA was weakly correlated with BMI, TG, hs-CRP, and eGFR, while in women it was additionally correlated with HDL cholesterol and in men, with FPG. Correlation with vascular parameters was very weak. 

### 3.3. Linear Regression Analysis of cfPWV and SUA

We conducted a sex-divided multiple linear regression analysis (Table 6), with the lowest SUA (Q1) as a reference group. It showed that, in women, the third (Q3) and fourth (Q4) quartiles of SUA were significantly associated with cfPWV after multiple adjustments for age, BMI, hs-CRP, hemodynamic parameters, and kidney function. On the other hand, in men, an association was present only in the fourth (Q4) quartile when adjusted for age, BMI, and eGFR and was lost after further adjustments.

## 4. Discussion

Early vascular aging is an emerging issue that must be addressed since it eventually affects the whole arterial tree and causes a higher risk for premature cardiovascular morbidity and mortality. In our study, we confirmed that, in a population-based, middle-aged Lithuanian cohort with metabolic syndrome, even “normal” levels of serum uric acid are associated with inflammation, higher mean arterial pressure, and stiffer arteries measured by aortic stiffness. Indeed, a greater carotid-femoral pulse wave velocity following multiple adjustments was associated with higher serum uric acid levels in women but not in men. Since the median cfPWV in women was 8.6 m/s, reflecting rather low cardiovascular risk, we suggest that hyperuricemia in this certain population indicates an early proinflammatory phenotype that contributes to EVA.

In our study, SUA levels were significantly associated with central arterial stiffness (cfPWV) in women. In men, the correlation was present and persisted after adjustment for age, BMI, and eGFR. However, it was lost after adjustments with other conventional risk factors such as hs-CRP and MAP. The relationship between hyperuricemia and arterial stiffening is still controversial since various studies share heterogeneous results. A systematic review and meta-analysis by Rebora et al. [24] revealed that in the general population the relationship between SUA and cfPWV exists with low-moderate heterogeneity of the six studies (I^2^ = 34%), being significant in both males and females (β: 0.06, 95%CI: 0.03, 0.09, and β: 0.07, 95%CI: 0.03, 0.11, respectively). However, in the hypertensive population, only one study out of four succeeded in finding a significant association between SUA and aortic stiffness after adjustments for risk factors. Indeed, there were two other studies that lost the significance in adjusted analysis. A large Chinese study, involving normotensive and hypertensive patients, revealed that, although the relationship of hyperuricemia and cfPWV was present in both groups, in normotensives, a higher SUA tended to have more influence and accelerated the augmentation of cfPWV, in comparison to the hypertension group where the cfPWV was higher and continued to grow slightly [25]. Thus, we may hypothesize that, having lower MAP in the female group in this study, the impact of hyperuricemia may be more expressed. 

In our study no relationship at all in both sexes was observed regarding peripheral arterial stiffness, measured by crPWV. Liang et al. also evaluated the relationship between SUA and crPWV, which failed to remain significant after adjustment for heart rate, blood pressure, and lipid profile [25]. Similarly, in a study by Bian et al., there was no difference between crPWV in hyperuricemic or normouricemic groups in both sexes [26]. Bian et al. presumed that SUA has an impact only on large arterial stiffening, measured by cfPWV, while peripheral arteries do not respond to hyperuricemia [26]. Our findings sustain this thought. An elastic artery, such as the aorta, is more vulnerable to inflammation that leads to the elastin degradation and its replacement by collagen [27].

The sex specific differences in relationship between SUA and arterial stiffness might be determined by sex hormones since estrogens have an uricosuric effect [28]. As for metabolic syndrome, it was believed that estrogens protect and inhibit the development of metabolic syndrome in premenopausal women. However, hormone replacement therapy could not show a beneficial outcome in components of metabolic syndrome, suggesting that other sex hormones might have a more significant role [29,30]. However, the decrease in estrogens is known to affect nitric oxide (NO) production leading to vascular stiffening [31]. Additionally, according to the guidelines on cardiovascular disease prevention in clinical practice, early menopause is also considered a risk factor for CVD [19]. In this study, we found a sex-dependent relationship between SUA and vascular parameters, suggesting the importance of hormones in EVA pathogenesis. Bian et al. also reported a correlation between SUA and cfPWV in women after adjusting for conventional risk factors. To note, their study also included younger men and women, and hypertension and smoking were more prevalent among men [26]. In our study, there were also more than twice as many smoking men as women, although no difference between smokers’ and non-smokers’ groups in respect to SUA was present. 

The reason why a relationship between arterial parameters and SUA differs between sexes might be influenced by the age range of participants and different methods of evaluating arterial stiffness, as well as the inclusion of diverse risk factors for CVD. In the present study, women had higher BMI and LDL cholesterol and poorer kidney function. Men, however, showed lower HDL cholesterol, higher TG and MAP, and a higher prevalence of smokers. Of course, the age discrepancy in our cohort should be also taken into account. Age is known to be one of the most important factors for arterial stiffening [13,32]. That may explain why the relationship between cfPWV and hyperuricemia in this study was present only in women who were older than men. Furthermore, Mehta et al. [33] in the Framingham study suggested an idea that hyperuricemia induces arterial stiffening in individuals who already have an elevated risk for CVD. Moreover, inflammation was less expressed in males and, despite presenting higher hyperuricemia, they had better vascular health as compared to women.

While cfPWV is a gold standard for measuring arterial function, CIMT, on the other hand, mainly represents the structure of the carotid artery [13]. Studies have shown a large age impact on the relationship between SUA and CIMT [34,35,36]. Nevertheless, in this study we had a different age range between the men and women groups; however, we did not succeed in finding an association in either of them.

Another arterial parameter, FMD, which is used to determine an endothelial dysfunction, showed no relationship with SUA in our study. Only several small studies have been performed on this topic, and the findings are controversial [37,38,39]. Some authors hypothesized that FMD may be an early predictor of vascular dysfunction, and including patients with coronary artery disease and/or diabetes may lead to a diminished difference between normouricemic and hyperuricemic patients [40]. In the present study, as the patients had metabolic syndrome, it may have lowered the role of SUA on FMD as well. 

In our study we also showed that hyperuricemia was closely associated with inflammation, measured by hs-CRP. Similarly, Tsai et al. [41] in a study of 200 essential hypertension patients showed a significant association between SUA and CRP, although hs-CRP was not associated with crPWV. On the other hand, another study carried out in Japan reported that both elevated SUA and hs-CRP related with vascular stiffness, measured by brachial-ankle pulse wave velocity (baPWV) [42]. Since crPWV measures arterial stiffness in muscular arteries and baPWV does so both in elastic and muscular arteries, the results are inconsistent. Meanwhile, cfPWV, used in our study, reflects large artery remodeling and, therefore, could reveal that a more expressed proinflammatory phenotype in middle-aged women with metabolic syndrome alongside higher MAP if hyperuricemia is present might indicate sex-specific pathways of early vascular aging. Interestingly, according to the Guidelines on cardiovascular disease prevention in clinical practice released in 2021, anti-inflammatory therapy with a low dose of colchicine is suggested only for high-risk patients as a secondary prevention [19]. It would be interesting to see if colchicine is more effective in women with metabolic syndrome as a primary prevention.

### Limitations

Increased circulating SUA is the consequence of a high intake of purine-rich products, increased SUA production, decreased renal elimination, or a combination of these mechanisms. Our limitation is that we did not evaluate the intake of SUA with food and alcohol in this study. Additionally, the absence of information on which an antihypertensive medication group was used in study subjects might have influenced results because both beta-blockers and diuretics can increase uricemia. Finally, we did not exclude subjects with pathologies, such as autoimmune disease or neoplasia, which might have induced their SUA concentration. 

## 5. Conclusions

The findings of our study support that hyperuricemia leads to a proinflammatory state, measured by hs-CRP. We found no relationship between SUA and CIMT, QCS, crPWV, and FMD. However, we provide evidence that high serum uric acid, in combination with inflammation, is associated with arterial stiffness, measured by carotid-femoral pulse wave velocity, in women in a population-based, middle-aged Lithuanian cohort with metabolic syndrome. 

## Figures and Tables

**Figure 1 medicina-58-00373-f001:**
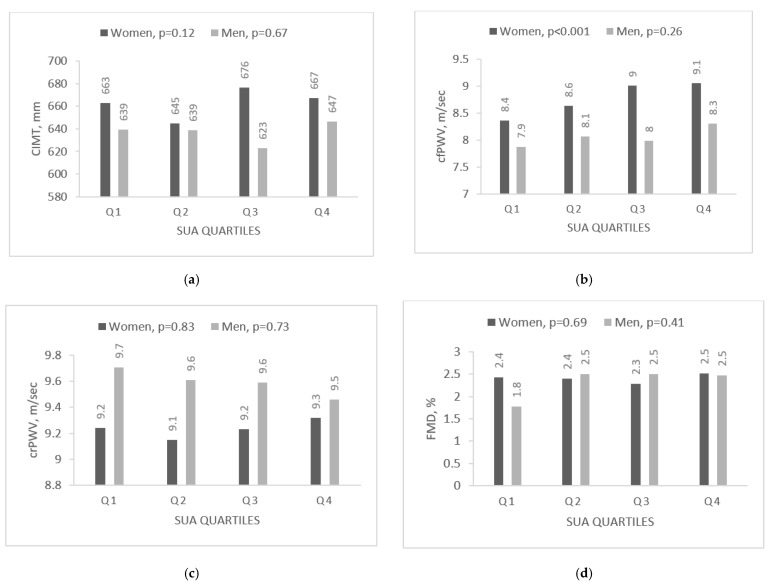
The comparison of arterial parameters by sex according to uric acid quartile: (**a**–**e**). CIMT, carotid intima-media thickness; cfPWV, carotid-femoral pulse wave velocity; crPWV, carotid-radial pulse wave velocity; QCS, quality carotid stiffness; SUA, serum uric acid.

**Table 1 medicina-58-00373-t001:** Descriptive characteristics by sex.

Characteristics	Total (*n* = 696)	Women (*n* = 439)	Men (*n* = 257)	*p*-Value
Age, year	54 ± 7	58 ± 4	47 ± 4	<0.001
BMI, kg/m^2^	31.6 ± 4.5	31.9 ± 4.9	31.2 ± 3.7	0.03
FPG, mmol/L	6.1 (3.4; 23.5)	6.5 (3.4; 23.5)	6.4 (4.8; 18.1)	0.255
TC, mmol/L	6.04 ± 1.36	6.11 ± 1.35	5.91 ± 1.34	0.05
LDL cholesterol, mmol/L	3.81 ± 1.16	3.88 ± 1.15	3.69 ± 1.16	0.04
HDL cholesterol, mmol/L	1.25 ± 0.32	1.36 ± 0.31	1.08 ± 0.24	<0.001
TG, mmol/L	1.75 (0.48; 32.05)	1.66 (0.48; 18.4)	2.0 (0.49; 32.05)	<0.001
MAP, mmHg	100 ± 10	99 ± 10	102 ± 10	<0.001
SUA, µmol/L	358.5 ± 86.38	333.51 ± 81.66	401.27 ± 78.24	<0.001
eGFR, mL/min/1.73 m^2^	91 ± 12	88 ± 11	97 ± 11	<0.001
hs-CRP, mg/L	1.6 (0.13; 47.1)	1.67 (0.18; 47.1)	1.53 (0.13; 42.9)	0.189
Hyperuricemia, % (*n*)	34 (234)	35 (154)	32 (85)	0.464
Gout, % (*n*)	2 (17)	1 (6)	4 (11)	0.016
Smoking, % (*n*)	26 (179)	17 (75)	41 (104)	<0.001

BMI, body mass index; FPG, fasting plasma glucose; TC, total cholesterol; LDL, low-density lipoprotein; HDL, high-density lipoprotein; TG, triglycerides; MAP, mean arterial pressure; SUA, serum uric acid; eGFR, estimated glomerular filtration rate; hs-CRP, high-sensitivity C-reactive protein.

**Table 2 medicina-58-00373-t002:** Arterial parameters by sex.

Characteristics	Total (*n* = 696)	Women (*n* = 439)	Men (*n* = 257)	*p*-Value
QCS (right)	4.1 (1.3; 17.7)	4.6 (1.3; 17.7)	3.6 (1.5; 16.7)	<0.001
QCS (left)	4.3 (1.1; 15.4)	4.8 (1.4; 15.4)	3.6 (1.1; 14)	<0.001
cfPWV, m/sec	8.5 ± 1.45	8.76 ± 1.49	8.06 ± 1.26	<0.001
crPWV, m/sec	9.4 ± 1.2	9.23 ± 1.15	9.6 ± 1.25	<0.001
CIMT (mean of right and left), μm	653 ± 103	663 ± 99	637 ± 108	0.002
FMD, %	2.39 (0.17; 15.44)	2.4 (0.2; 15.44)	2.35 (0.17; 9.82)	0.15

QCS, quality carotid stiffness; cfPWV, carotid-femoral pulse wave velocity; crPWV, carotid-radial pulse wave velocity; CIMT, carotid intima-media thickness, FMD, flow-mediated dilatation.

**Table 3 medicina-58-00373-t003:** The comparison of objective data according to uric acid quartile in women.

Characteristics	Women (*n* = 439)				*p*-Value
Quartiles	Q1	Q2	Q3	Q4	
Number	112	111	108	108	
SUA, µmol/L	≤277	278–326	327–381	≥382	
Age, year	58 ± 4	57 ± 4	58 ± 4	57± 4	0.946
BMI, kg/m^2^	30.2 ± 4.6	30.7 ± 4.4	32.9 ± 4.7	33.9 ± 5	<0.001
FPG, mmol/L	5.9 (3.4; 16.8)	6.1 (5; 23.5)	6 (4.6; 12.1)	6.3 (5; 16.1)	0.014
TC, mmol/L	6.16 ± 1.43	5.95 ± 1.32	6.15 ± 1.23	6.18 ± 1.44	0.532
LDL cholesterol, mmol/L	3.92 ± 1.22	3.79 ± 1.16	3.99 ± 1.0	3.82 ± 1.21	0.585
HDL cholesterol, mmol/L	1.46 ± 0.34	1.35 ± 0.28	1.34 ± 0.25	1.29 ± 0.33	<0.001
TG, mmol/L	1.51 (0.48; 8.58)	1.50 (0.52; 18.40)	1.68 (0.64; 5.06)	1.93 (0.58; 7.3)	<0.001
MAP, mmHg	98 ± 10	98 ± 10	101 ± 11	101 ± 10	0.032
eGFR, mL/min/1.73 m^2^	92.1 ± 8.9	88.8 ± 8.6	86.3 ± 10.4	84.2 ± 13	<0.001
hs-CRP, mg/L	1.37 (0.18; 28.4)	1.34 (0.19; 47.1)	1.98 (0.33; 18.1)	2.75 (0.33; 25.3)	<0.001
Smoking, % (*n*)	21 (24)	14 (15)	17 (18)	17 (18)	0.472

BMI, body mass index; FPG, fasting plasma glucose; TC, total cholesterol; LDL, low-density lipoprotein; HDL, high-density lipoprotein; TG, triglycerides; MAP, mean arterial pressure; SUA, serum uric acid; eGFR, estimated glomerular filtration rate; hs-CRP, high-sensitivity C-reactive protein.

**Table 4 medicina-58-00373-t004:** The comparison of objective data according to uric acid quartile in men.

Characteristics	Men (*n* = 257)				*p*-Value
Quartiles	Q1	Q2	Q3	Q4	
Number	65	67	61	64	
SUA, µmol/L	≤352	353–393	394–452	≥453	
Age, year	49 ± 4	48 ± 4	46 ± 4	46 ± 4	<0.001
BMI, kg/m^2^	30.2 ± 3.1	30.2 ± 3.5	31.7 ± 3.4	32.7 ± 4.1	<0.001
FPG, mmol/L	6.05 (5.03; 17.54)	6.04 (4.83; 18.07)	5.94 (4.93; 7.78)	6.08 (4.9; 11.49)	0.151
TC, mmol/L	5.57 ± 1.36	6.08 ± 1.38	6 ± 1.17	5.98 ± 1.39	0.132
LDL cholesterol, mmol/L	3.46 ± 1.2	3.83 ± 1.13	3.74 ± 1.06	3.74 ±1.23	0.312
HDL cholesterol, mmol/L	1.06 ± 0.25	1.15 ± 0.27	1.05 ±0.2	1.05 ± 0.24	0.039
TG, mmol/L	1.65 (0.49; 32.05)	1.78 (0.53; 6.93)	2.23 (0.96; 6.61)	2.51 (0.79; 7.8)	<0.001
MAP, mmHg	101 ± 9	100 ± 10	101 ± 10	105 ± 11	0.057
eGFR, mL/min/1.73 m^2^	99.8 ± 10.6	97.3 ± 9.7	98.7 ± 11.3	94.3 ± 12.7	0.03
hs-CRP, mg/L	1.26 (0.2; 27.3)	1.37 (0.13; 10.3)	1.94 (0.24; 42.9)	1.89 (0.47; 31.8)	<0.001
Smoking, % (*n*)	38 (25)	46 (31)	38 (23)	39 (25)	0.752

BMI, body mass index; FPG, fasting plasma glucose; TC, total cholesterol; LDL, low-density lipoprotein; HDL, high-density lipoprotein; TG, triglycerides; MAP, mean arterial pressure; SUA, serum uric acid; eGFR, estimated glomerular filtration rate; hs-CRP, high-sensitivity C-reactive protein.

**Table 5 medicina-58-00373-t005:** The correlation between objective data, vascular parameters and SUA by sex.

Characteristics	Women (*n* = 439)		Men (*n* = 257)	
	Spearman’s Correlation Coefficient r	*p*-Value	Spearman’s Correlation Coefficient r	*p*-Value
Age, year	−0.02	0.68	−0.26	<0.001
BMI, kg/m^2^	0.32	<0.001	0.3	<0.001
FPG, mmol/L	−0.06	0.35	0.12	0.012
TC, mmol/L	0.01	0.853	0.1	0.102
LDL cholesterol, mmol/L	−0.02	0.623	0.06	0.34
HDL cholesterol, mmol/L	−0.23	<0.001	−0.07	<0.27
TG, mmol/L	0.22	<0.001	0.27	<0.001
MAP, mmHg	0.14	0.003	0.14	0.02
eGFR, mL/min/1.73 m^2^	−0.23	<0.001	−0.14	0.03
hs-CRP, mg/L	0.29	<0.001	0.27	<0.001
QCS (right)	0.13	0.008	−0.05	0.39
QCS (left)	0.11	0.02	−0.02	0.8
cfPWV, m/sec	0.19	<0.001	0.13	0.03
crPWV, m/sec	0.02	0.78	−0.04	0.56
CIMT (mean of right and left), μm	0.05	0.34	0.02	0.81
FMD, %	0.05	0.33	0.08	0.121

BMI, body mass index; FPG, fasting plasma glucose; TC, total cholesterol; LDL, low density lipoprotein; HDL, high density lipoprotein; TG, triglycerides; MAP, mean arterial pressure; SUA, serum uric acid; eGFR, estimated glomerular filtration rate; hs-CRP, high-sensitivity C-reactive protein; QCS, quality carotid stiffness; cfPWV, carotid-femoral pulse wave velocity; crPWV, carotid-radial pulse wave velocity; CIMT, carotid intima-media thickness; FMD, flow-mediated dilatation.

**Table 6 medicina-58-00373-t006:** Multiple linear regression analysis with cfPWV as dependent variable.

Women				Men		
Estimate	B	SE	*p*-Value	B	SE	*p*-Value
Model 1						
Q2 uric acid	0.213	0.187	0.257	0.259	0.219	0.238
Q3 uric acid	0.448	0.195	0.022	0.17	0.231	0.462
Q4 uric acid	0.468	0.203	0.021	0.478	0.242	0.049
Model 2						
Q2 uric acid	0.214	0.182	0.238	0.211	0.209	0.312
Q3 uric acid	0.38	0.19	0.047	0.138	0.221	0.535
Q4 uric acid	0.459	0.198	0.021	0.34	0.234	0.147
Model 3						
Q2 uric acid	0.257	0.181	0.156	0.209	0.209	0.317
Q3 uric acid	0.384	0.189	0.042	0.144	0.222	0.518
Q4 uric acid	0.491	0.197	0.013	0.328	0.234	0.162
Model 4						
Q2 uric acid	0.216	0.18	0.23	0.288	0.213	0.179
Q3 uric acid	0.397	0.186	0.033	0.263	0.227	0.249
Q4 uric acid	0.533	0.194	0.006	0.415	0.24	0.085

Q1 SUA as a reference group; Model 1: adjusted for age, BMI, and eGFR; Model 2: Model 1 + hs-CRP and MAP; Model 3: Model 2 + CIMT; Model 4: Model 3 + FMD.

## Data Availability

Not applicable.

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
