# Peer review of "Inflammaging and Vascular Function in Metabolic Syndrome: The Role of Hyperuricemia"

_medicina, 2022, doi:10.3390/medicina58030373_

Round 1
Reviewer 1 Report
I find the article topic of interest and presented in a clear manner. However, I consider that there are some aspects that require qualification.
- The studied group includes patients with gout. What are the reasons why these patients were not excluded?
Patients with gout were not on specific treatment?
2. The division of patients into quartiles is not explained either in terms of the justification of this approach or the range of values used.
The result may be biased because in Q3 there are patients with normal SUA and hyperuricemia.
3.There is a need to better define the studied population. Were patients with different kind of neoplasia and autoimune diseases excluded?
Author Response
Response to Reviewer 1 Comments
We are very grateful for Your reviews that helped to improve our paper. Please see below in red our responses to your comments.
Point 1: The studied group includes patients with gout. What are the reasons why these patients were not excluded? Patients with gout were not on specific treatment?
Response 1: You have raised an important point here, as we have left some patients with diagnosed gout. However, we excluded every subject, who declared the usage of xanthine oxidase inhibitors. Rest of the subjects who were diagnosed with gout did not declare using medicine. We added this information to the manuscript. We agree, that next time it should be monitored more precisely as the percentage of the subjects with gout and without treatment is rather high. (lines 90-91)
Point 2: The division of patients into quartiles is not explained either in terms of the justification of this approach or the range of values used.
The result may be biased because in Q3 there are patients with normal SUA and hyperuricemia.
Response 2: We have divided subjects into the quartiles using R-Commander. You are right that subjects in the Q3 are either with normal SUA or hyperuricaemia, however they all have rather high serum uric acid concentration which is near the upper normal limit or slightly over it. Our aim by this division was to make a more detailed analysis between SUA concentration and parameters, instead of dividing subjects only to the normouricaemic and hyperuricaemic groups. We added additional information in the description to make this division more clear. (lines 153-155)
Point 3: There is a need to better define the studied population. Were patients with different kind of neoplasia and autoimune diseases excluded?
Response 3: Thank you for pointing this out. Unfortunately, we had no information about these conditions, however we understand its importance on serum uric acid concentration. Therefore, we added a comment to the limtations section. (lines 329-331)
Reviewer 2 Report
Agnė Laučytė-Cibulskienė and Colleagues presented a very interesting and valuable research paper. The purpose of this paper was to analyze the relationship between serum uric acid concentration and some anthropometric, biochemical, epidemiological and vascular parameters. The methodology used in the study is at high level. Results are well presented. The subject of the study is very important because metabolic syndrome and its consequences (first of all cardiovascular complications) is one of the crucial problems for public health worldwide. Better understanding of pathogenesis of the metabolic syndrome may lead to availability of more and more effective methods of diagnosis and treatment.
However, I believe that some changes to the manuscript are needed which may further improve the quality and relevance of the text. In brackets, I give the numbers of lines to which the comment relates.
I think, the number of references should be given rather in “[ ]” than in “( )”.
The term „inflammaging” should be explained more precisely, because it may be not understandable for some readers. The Authors have shown some examples what the theory of inflammaging explained, but they have not defined what exactly inflammaging is. (38-41)
Consider carefully, whether “(flow mediated dilatation, FMD)” would be more elegant than “(flow mediated dilatation (FMD))”. (61)
In my opinion the introduction should be significantly reorganized. The purpose of the study was not described. According to me it would be worth to divided the introduction into subsections, for example the first referring to early vascular aging and inflammaging, the second referring to the most important information associated with metabolic syndrome, and the third to describe the purpose of the study. The Authors mentioned the role of uric acid in the context of oxidative stress, but something more should be mentioned about the role of oxidative stress in the pathogenesis of metabolic syndrome and its complications. Recent findings indicating that obesity and insulin resistance are most strongly associated with oxidative stress in the course of metabolic syndrome are worth to mention.
It should be mentioned in which institution the Ethical Committee works – in university, in hospital? (65-66)
Explain please why the value of 100 mmHg inflation of a pneumatic tourniquet has been chosen and why the second measurement of diameter has been performed exactly 3 minutes after deflation. The value of inflation of pneumatic tourniquet was the same (100 mmHg) for all patients not depending on the value of patients systolic blood pressure directly before the performance of FMD procedure?
In table 1 it should be “SUA, mmol/l”, as well as “hs-CRP, mg/l”.
In all tables it should be “Men (n=257)” (lack of n in the current form).
The subtitles “General and vascular parameters in respect of hyperuricaemia in metabolic syndrome” (158) and “Linear regression analysis: uric acids influence on carotid-femoral pulse wave velocity” (184) should be written in the format appropriate for subtitles.
Table 3 is difficult to read, especially positions for which there is too little place in one row (FPG, hs-CRP, TG). Maybe it would be better to prepare separate tables for women and for men?
Conclusions should be more precisely described. The current form is laconic.
The reference is prepared not in accordance to the rules required in papers published by MDPI.
English seems to be quite good (the text is understandable for me, but I’m not English philologist and I feel not fully able to assess language quality).
Author Response
Response to Reviewer 2 Comments
We are very grateful for Your reviews that helped to improve our paper. Please see below in red our responses to your comments.
Point 1: I think, the number of references should be given rather in “[ ]” than in “( )”.
Response 1: Thank you for the notice, it is now corrected in the manuscript.
Point 2: The term „inflammaging” should be explained more precisely, because it may be not understandable for some readers. The Authors have shown some examples what the theory of inflammaging explained, but they have not defined what exactly inflammaging is. (38-41)
Response 2: We agree with this suggestion, therefore we added additional information about inflammaging. (lines 38-39)
Point 3: Consider carefully, whether “(flow mediated dilatation, FMD)” would be more elegant than “(flow mediated dilatation (FMD))”. (61)
Response 3: We agree that double bracket should be avoided, therefore we slightly changed the sentence. (line 74)
Point 4: In my opinion the introduction should be significantly reorganized. The purpose of the study was not described. According to me it would be worth to divided the introduction into subsections, for example the first referring to early vascular aging and inflammaging, the second referring to the most important information associated with metabolic syndrome, and the third to describe the purpose of the study. The Authors mentioned the role of uric acid in the context of oxidative stress, but something more should be mentioned about the role of oxidative stress in the pathogenesis of metabolic syndrome and its complications. Recent findings indicating that obesity and insulin resistance are most strongly associated with oxidative stress in the course of metabolic syndrome are worth to mention.
Response 4: We agree with your suggestion that introduction part could be improved, therefore we made some changes to structurize it more clearly. Please find corrections in the manuscript. All changes are tracked in a Word file. (lines 32-76)
Point 5: It should be mentioned in which institution the Ethical Committee works – in university, in hospital? (65-66)
Response 5: Thank you for noticing that, we have added additional information. (lines 81-82)
Point 6: Explain please why the value of 100 mmHg inflation of a pneumatic tourniquet has been chosen and why the second measurement of diameter has been performed exactly 3 minutes after deflation. The value of inflation of pneumatic tourniquet was the same (100 mmHg) for all patients not depending on the value of patients systolic blood pressure directly before the performance of FMD procedure?
Response 6: We agree that the method of flow mediated dilatation measurement was poorly described, as well as there were some errors in it, therefore we added additional information to the description. Thank you for noticing that. (lines 130-132 and 135-139)
Point 7: In table 1 it should be “SUA, mmol/l”, as well as “hs-CRP, mg/l”.
Response 7: Thank you for the notice, it is now corrected in the manuscript.
Point 8: In all tables it should be “Men (n=257)” (lack of n in the current form).
Response 8: Thank you for the notice, it is now corrected in the manuscript.
Point 9: The subtitles “General and vascular parameters in respect of hyperuricaemia in metabolic syndrome” (158) and “Linear regression analysis: uric acids influence on carotid-femoral pulse wave velocity” (184) should be written in the format appropriate for subtitles.
Response 9: We agree with this suggestion, therefore we changed the subtitles into more appropriate forms. (lines 182 and 218)
Point 10: Table 3 is difficult to read, especially positions for which there is too little place in one row (FPG, hs-CRP, TG). Maybe it would be better to prepare separate tables for women and for men?
Response 10: Thank you for this suggestion. We splitted Table 3 into two parts as you have recommended.
Point 11: Conclusions should be more precisely described. The current form is laconic.
Response 11: We agree that our conclusions were not indeed expanded, therefore we added some information to them. However, we believe that in coclusions we wanted to ephasize the major finding of our study which is that “high serum uric acid, in combination with inflammation are associated with arterial stiffness, measured by carotid-femoral pulse wave velocity, in women in popula-tion-based middle-aged Lithuanian cohort with metabolic syndrome”. (lines 333-336)
Point 12: The reference is prepared not in accordance to the rules required in papers published by MDPI.
Response 12: Thank you for the notice, it is now corrected in the manuscript.
Round 2
Reviewer 2 Report
The manuscript has been significantly improved.
I suggest only to consider to use additional references in the introduction where described oxidative stress in the context of metabolic syndrome and insulin resistance (for example recent advances in this area are described in papers such as doi.org/10.3390/antiox11010079; doi.org/10.3390/antiox10071018).
Author Response
Thank you very much for your positive assessment of our modification of the text and recommended articles. We included them to improve our introduction part. (lines 45-47 and 50-54)